# The Face of Early Cognitive Decline? Shape and Asymmetry Predict Choice Reaction Time Independent of Age, Diet or Exercise

**William M. Brown *** and **Agnese Usacka**

School of Psychology, University of Bedfordshire, Park Square, Luton LU1 3JU, UK; agnese.ushacka@gmail.com
* Correspondence: william.brown@beds.ac.uk; Tel.: +44-(0)1582-74-3470

**Abstract:** Slower reaction time is a measure of cognitive decline and can occur as early as 24 years of age. We are interested if developmental stability predicts cognitive performance independent of age and lifestyle (e.g., diet and exercise). Developmental stability is the latent capacity to buffer ontogenetic stressors and is measured by low fluctuating asymmetry (FA). FA is random—with respect to the largest side—departures from perfect morphological symmetry. The degree of asymmetry has been associated with physical fitness, morbidity, and mortality in many species, including humans. We expected that low FA (independent of age, diet and exercise) will predict faster choice reaction time (i.e., correct keyboard responses to stimuli appearing in a random location on a computer monitor). Eighty-eight university students self-reported their fish product consumption, exercise, had their faces 3D scanned and cognitive performance measured. Unexpectedly, increased fish product consumption was associated with worsened choice reaction time. Facial asymmetry and multiple face shape variation parameters predicted slower choice reaction time independent of sex, age, diet or exercise. Future work should develop longitudinal interventions to minimize early cognitive decline among vulnerable people (e.g., those who have experienced ontogenetic stressors affecting optimal neurocognitive development).

**Keywords:** developmental stability; fluctuating asymmetry; face shape variation; geometric morphometrics; choice reaction time; diet; polyunsaturated fatty acids omega-3; physical activity early cognitive decline

## 1. Introduction

Cognitive performance declines with advancing age [1–4], possibly due to cumulative exposure to stress or biological deterioration (i.e., senescence). However, it may come as a surprise that cognitive deterioration can begin as early as 24 years of age [1]. One way to assess the degree of cognitive decline is speed of information-processing [3,4]. Multiple studies [3–6] have shown choice reaction time (CRT)—speed of accurately detecting and providing a motor response to one of four stimuli—is a sensitive measure of cognitive decline as we age. This is because CRT requires increased processing and sensorimotor integration to make a correct choice among possible responses [5,6]. It remains an open empirical question if the ability to buffer developmental stress causes some people to cognitively decline more slowly than others. Developmental stability often measured by departures from perfect morphological symmetry—is the latent capacity to buffer genetic or environmental stressors during ontogeny [7]. This study has been developed to determine if developmental stability predicts cognitive performance independent of age and lifestyle factors (e.g., diet or exercise).

### 1.1. Low Fluctuating Asymmetry: A Measure of Developmental Stability

Developmental stability [7,8] is often assessed by fewer departures in bilateral symmetry or the relative absence of fluctuating asymmetry (FA). FA is subtle size or shape differences between the left and right sides of bilateral traits (e.g., eyes, nostrils). Lower FA is moderately associated with better health outcomes [9] and has been shown to predict psychometric intelligence [8]. Similarly, faster simple reaction time (RT) predicts psychometric intelligence [10] and lower FA predicts faster CRT in children [11] as well as reduced age-related cognitive decline [3]. RT declines with age [3,12–14] predicting morbidity [15] and mortality [5,6]. Such findings have contributed to emergence of the field "cognitive epidemiology" and search for a latent factor of cognitive system integrity [2,4–6]. Given the pattern of previously reported associations between low FA and RT, some researchers have suggested that CRT and FA are both tapping into the underlying latent factor of developmental stability [3] or general system integrity.

In their landmark study, Penke et al. [3] measured symmetry from facial photographs and choice reaction time in older men. It was found that symmetrical men experienced less cognitive decline. Lower FA (controlled for directional asymmetry) predicted faster RT with a more profound effect on CRT performance. The argument made by Penke et al. [3] is that the shared variance is caused by a third latent variable, namely developmental integrity. Given the importance of understanding age-related cognitive decline for human health, we focused our investigation on whether facial asymmetry predicts cognitive performance. One problem with this hypothesis is that there are potentially confounded third variables (e.g., background and life-style factors such as age, diet and exercise that can affect both symmetry and cognitive performance). Indeed, it may be that lifestyle factors explain any association found between facial symmetry and enhanced CRT performance. Lifestyle affects the molecular epigenetic systems regulating development. Below we briefly introduce epigenetics with an emphasis on how lifestyle factors effect neurocognitive machinery and performance.

### 1.2. Do Healthy Lifestyles Protect Against Epigenetic Stress?

Epigenetics is the study of the molecular mechanisms of how we get from genotype to phenotype [16]. Specifically, from a modern molecular biological viewpoint, epigenetics is the study of heritable chemical or structural modifications regulating gene expression without making changes to the underlying DNA sequence [17]. The epigenome (a record of the chemical changes beyond or above the DNA sequence) is affected by pathogenic factors throughout the lifespan [18–20]. Environmental disruptions are a source of epigenetic stress and it is now well-accepted that departures from perfect symmetry are epigenetic in origin. FA is epigenetic because there is no genotype for the left and right side of a trait. Therefore, any size and shape differences are 'above the genome'. Epigenetic disruptions also affect neurocognitive development. For example, childhood trauma positively correlates with DNA methylation of the glucocorticoid receptor gene and associated with adult mood disorder [20]. Furthermore, epigenetic manipulations of brain chromatin can reverse abnormalities in animal models of neurodegenerative diseases, such as Alzheimer's disease (AD) [21]. In addition, DNA damage has shown to be a factor in ageing and longevity [22] and alterations in diet may protect against DNA damage [23]. Indeed, nutrition is a major factor in gene regulation [19,23]. Another effective epigenetic regulator that has the potential to neutralize pathophysiological insults (e.g., poor nutrition) is physical activity [24]. It is reasonable to presume that via epigenetic reprogramming, poor diet and physical inactivity can damage developmental or system integrity. Further, it is likely that diet and exercise benefits neurocognitive functioning and ameliorates cognitive decline as we age [4]. These possibilities are briefly detailed in the following sections.

1.2.1. Diet and Cognitive Performance

Polyunsaturated fatty acids omega-3 (n-3 PUFA), flavonoids, folic acid, and B group vitamins affect cognitive functioning [25–28]. Higher concentration of n-3 PUFA, specifically docosahexaenoic

acid (DHA), has been found to benefit brain development and improve cognitive performance [27–29]. A diet rich in n-3 PUFA decreases cognitive decline in healthy individuals [30,31] and to be effective in counteracting AD symptoms at the onset stage [26]. The underlying neural mechanisms possibly responsible for DHA-cognitive performance associations have been investigated. Neurons have high-energy demands and vitally depend on the oxidizing properties of phospholipids [32,33]. The most prevalent phospholipid in the brain is DHA [28] promoting homeostasis [34]. Indeed, lower DHA concentration in neuronal membrane impairs mitochondrial function harming neuronal metabolism and can lead to cell death [34]. It is known that poor diets can detrimentally impact the epigenome, but can physical exercise have similar beneficial effects on neurodevelopment and cognitive function [18,19]?

### 1.2.2. Exercise and Cognitive Performance

Like DHA, physical exercise has beneficial effects on neurocognitive function [35–41]. Exercise stimulates brain-derived neurotrophic factor (*BDNF*) inciting mitochondrial activation [32]. Further, exercise may regulate differentiation of neurons in developing brains [35], enhancing neural plasticity, and protecting from neurodegeneration [37]. High intensity exercise activates *BDNF* [35–37] and has been shown to improve memory, attention and processing speed [36]. Cycling also improves selective attention and processing speed [37], whilst rope skipping and running affected executive function and academic achievement in overweight children [38]. Rowing can improve memory performance in sedentary elderly individuals [39]. Interestingly, physical activity can prevent or reverse age-related cognitive decline and has become one of the most widely recommended non-pharmacological interventions in reversing cognitive impairment amongst AD patients [42]. Regular physical exercise changes *BDNF* DNA methylation and gene expression promoting neuroplasticity [43–45].

It is clear based on previous work, that diet and exercise shapes epigenomes and neurocognitive function [18,25–48]. However, it remains unclear if some people are better at buffering developmental stress than others. The general premise here is that more symmetrical people may be better developmentally equipped to buffer environmental stressors. However, are these associations independent of age, diet or exercise? Here, we explore these ideas further in a cross-sectional correlational study of facial symmetry and cognitive performance.

### 1.3. Current Study: Developmental Stability, Cognitive Performance and Controlling for Lifestyle and Background Factors

The premise of this paper is that developmental stability predicts better cognitive performance. However, are the effects of developmental stability independent of age, diet or exercise? Specifically, beyond neurocognitive effects, lifestyle factors and ageing affects morphology (e.g., fat and muscle distribution). Facial morphology is affected by lifestyle factors and ageing [49–51], which should bring caution when interpreting any relation between facial morphology and cognitive performance. Since faces are built, in part, of soft tissue, there is substantial change across the life course. Perhaps one of the most striking demonstrations of these developmental effects is when non-shared environments differentially affect monozygotic twins' facial appearance [49–52]. For instance, it has been shown that smoking twins compared to their non-smoking counterparts had several distinct facial attributes, such as lower lid bags, more pronounced nasolabial folds and upper lip wrinkles, as well as periorbital premature ageing [50]. Smoking has also been shown to increase asymmetry of the occlusion plane and upper eyelid ptosis (relative to a non-smoking twin). If a twin regularly slept in a prone position and uses dentures, they have greater nasal asymmetries than the twin that does not [51]. Beyond the effects of unhealthy lifestyles (e.g., smoking) on facial morphology, facial structure also changes as we age. Specifically, age decreases tissue elasticity, redistribution of subcutaneous fullness [53], and affects the bony structure of the face [54]. Shaw et al. [54] compared tomographic scans of young, middle aged and old individual facial skeletons to examine the ageing of the bone. They found an age-related difference

in orbital aperture area and its width, decreased bone volume in the midface area with widening of the pyriform aperture region and an increased angle of the jawbone in older individuals [54].

Beyond the effects of ageing on facial appearance, degree of facial asymmetry in early life may predict outcomes in later life [55–58]. Early chronic stress disrupts cognitive performance [59] and its effects exacerbate as we age [3,13,14]. We suspect that early chronic stress affects degree of facial asymmetry. Indeed, previous work [3] has shown that less symmetrical children have higher CRT scores [11]. Such findings are a concern as slower reaction time predicts increased risk of premature death and cardiovascular disease independent of age [60].

### 1.4. Hypotheses

In line with the cognitive epidemiology literature [3,4], we used CRT to isolate individual differences in cognitive performance. If facial FA indicates the quality of individual's phenotype [55] and CRT indicates a brain's efficiency processing information [10] then these measures are expected to correlate even though they measure different aspects of underlying system integrity (i.e., physical vs cognitive). Therefore, we hypothesized that developmental stability (measured by lower scores on facial fluctuating asymmetry) independent of sex, age, physical activity, and a diet rich in n-3 PUFA (found in fish products) will predict enhanced CRT performance (lower CRT scores indicate faster responses). Other sources of face shape variation beyond symmetry were included in the model since as data-driven sources of morphological variation may account for more variance than traditional theoretical-driven measures (e.g., FA). We also, expect that individuals, independent of age, who are physically active and consume a diet rich in n-3 PUFA will display lower FA.

## 2. Materials and Methods

### 2.1. Design

A correlational design was tested the hypothesis that developmental stability (i.e., low facial FA) predicts cognitive performance independent of sex, age, diet or exercise. We also included data-driven principal components of face shape variation as these may be better predictors than more theoretically driven measures such as FA.

### 2.2. Participants

We followed the Declaration of Helsinki and the British Psychological Society Ethical Guidelines (Institutional ethical approval certificate number WBLW16128041819). Ninety-two university students were opportunity sampled around campus (which included a participant recruitment pool electronic notice board). We were constrained to conduct this work on campus as the delicate Artec MH 3D scanner (Artec 3D, Luxembourg) is tethered to an expensive computer. Four participants' data could not be used due to missing data or 3D scan technical difficulties. The remaining 88 (47 female and 41 males) participants were aged 18 to 55 years ($M = 25.74$; $SD = 9.26$).

### 2.3. Materials

Fish product intake. To assess self-reported consumption of fish products (higher fish consumption indicates higher n-3 PUFA intake), Food Frequency Questionnaire (FFQ) measuring seafood intake [61,62] was used. It consists of nine items questioning the frequency of fish as a main course, fish as a bread spread or as a starter and fish oil consumption over the last 12 months, the type of fish or fish oil consumed and the amount. A moderate construct validity was identified ($r = 0.6$) among young adults in Europe against a 14-day food record [62]. It was identified to indicate higher n-3 PUFA with higher intake of fish products [63]. In the current study the quantity of various fish products (e.g., fish as starter, fish as main dish or fish oil supplement) showed reasonable internal consistency (Cronbach's Alpha = 0.67) and was within the range of previous work [63]. Participants responded using ordinal scales categorized into three domains: fish consumed as (1) starter, (2) main or

(3) fish consumed in oil form. Participants ordinal responses ranged from 1–3 in each category of fish consumption—1 = "low consumers (≤2 times per month)", 2 = moderate consumer (>2 to <8 times per month)" and 3= "high consumers (≥8 times per month)" as instructed by Thorsdottir, Birgisdottir, Kiely, Martinez and Bandarra [63]. Participants' responses in these three domains were summed to make a fish product consumption overall frequency score ranging from 3–9. Higher scores indicated higher fish product intake. It is important to note that Birgisdottir et al. [62] found the FFQ may overestimate fish intake despite being able to accurately detect individual differences in fish consumption.

Physical activity intensity measure. World Health Organization's Global Physical Activity Questionnaire (GPAQ) [64] was used to assess participants' physical activity intensity. The GPAQ contains 16 items that cover three activity types—activity at work, travel to and from places, and recreational activities. All domains were classified into two levels of intensity–moderate and vigorous activity. It has been shown to have a low to moderately-high construct validity ($r = 0.25$ to $r = 0.63$) against measures of physical fitness, body composition, and objective measures (e.g., tpedometer and accelerometer measures) of physical activity and acceptable test-retest reliability—short term reliability (10 days) in a range of 0.83 to 0.96, long term (three months) in a range of 0.53 to 0.83 [65]. In our sample, we found reasonable internal consistency for GPAQ's measures of time spent exercising (in minutes) as Cronbach's Alpha was 0.76.

Collected responses underwent data cleaning procedures according to Armstrong and Bull [64] instructions. Participants reported on the number of days, hours and minutes being physically active for each domain and intensity. If a case reported implausible value, had inconsistent answers or exceeded the maximum value (>16 h per day), it was removed from analysis. Participants' responses were kept in the analysis if at least one response on a sub-domain was valid.

To calculate participants' physical activity per week, a total metabolic equivalent of task (MET) units was calculated for time reported being physically active. Physical activity expends energy and this energy can be described by a MET unit as energy expenditure of an activity—1 MET represents the rate of energy expenditure while at rest; 4 MET (moderate intensity activity) or 8 MET (vigorous intensity activity) units expends four and eight times the energy used by the body at rest [66]. For example, 4 MET activity done for 30 min or 8 MET done for 15 min translates into 120 MET-minutes ($4 \times 30$ or $8 \times 15$).

Minutes in moderate intensity activities for each domain (work, travel, recreational activities) were summed, multiplied by the days per week in activity and multiplied by 4 MET. The resulted number provided with a MET-minutes value. For example, 120 minutes × 3 days × 4 MET = 1440 MET-minutes of moderate physical activity in a week. Vigorous activity MET-minutes were calculated in the same manner. Lastly, MET-minutes for moderate physical activity and for vigorous activity were summed to compute the total physical activity MET-minutes value for each participant.

Morphological measurements. Each participants' facial morphology was captured using an Artec MH 3D scanner (Artec 3D, Luxembourg). Artec Studio software (Artec 3D, Luxembourg) reconstructed facial topography to create a 3D image, which was imported into Landmark (Institute for Data Analysis and Visualization, UC Davis, Davis, California, CA, USA) [67], which is software for placing the 12 landmarks (Figure 1) on specimens.

To reduce measurement error, two sets of measurements were performed. These measurements underwent reliability analysis and disclosed a perfect reliability for all landmark placements ($\alpha = 1$). Averaged landmarks coordinates (x, y, and z dimensions) were imported into MorphoJ software (University of Manchester, Manchester, UK) for geometric morphometric analyses [68]. A Procrustes fit was performed on the landmark coordinates to separate shape from size and orientation. The advantage of geometric morphometrics methodology is that it allows us to identify and quantify shape features [69] that are independent of specimen size or orientation. Beyond shape characteristics, faces also have object symmetry. To assess symmetry, the landmarks on one side of the face are reflected to the other side. The observed shape allows MorphoJ (University of Manchester, Manchester, UK) to compare the mirrored reflection to actual shape configuration of landmarks [70]. From the 12 landmarks x, y, z

dimensional positions, MorphoJ (University of Manchester, Manchester, UK) yields 36 asymmetric shape components. The variance among the asymmetry components provided the FA measure, while the mean value represents DA. Like FA, DA is bilateral trait deviation from a perfect symmetry. However, it is directionally orientated on one side and is normally not thought to be indicative of phenotypic quality and could be functional (e.g., in the case of the human face indicative of asymmetries in facial expression). In our case, DA was near zero and absolute values were leptokurtic indicating it may be tapping into developmental stability. Therefore, we combined our FA and DA measures into a measure of FA about the absolute DA, like previous work [70].

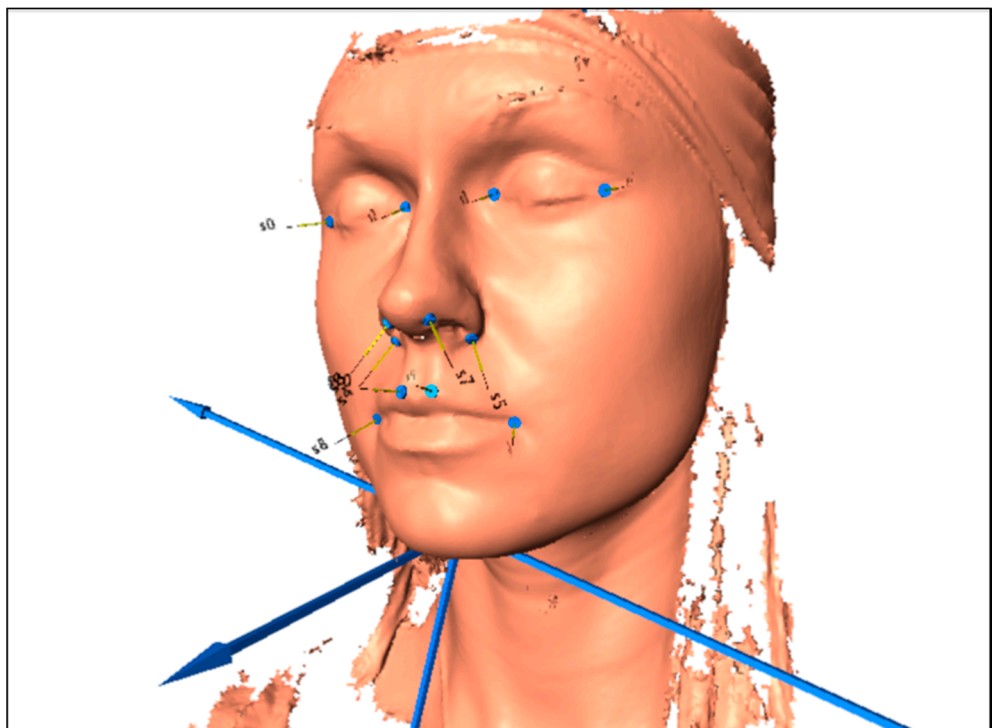

**Figure 1.** The location of the 12 landmarks placed on 3D face scans.

To measure facial morphology beyond asymmetry variation, MorphoJ (University of Manchester, Manchester, UK) [68] extracted facial shape components (PCs) of symmetrical shape variation, which accounted for 64.32 percent of the face shape variation. Each PC (Figure 2) accounted for 30.88, 19.33, 14.12 percent of the shape variance respectively. We included the face shape PCs in the analysis as data-driven face shape variation may be a better predictor of health and quality (e.g., attractiveness) than theory-driven measures such as facial symmetry [58,71–73].

Choice reaction time (CRT). Computer based Deary-Liewald four-choice reaction time task [74] assessed CRT. It consists of RT task that measures the time needed to react to a stimulus. It is a basic assessment of processing speed [75], in which response is required to the same stimulus (an 'x') emerging on the screen. The second part of the program is the CRT task that requires participants to make an appropriate response to the same stimulus emerging in different locations (Figure 3). Response time taken to react to a stimulus was measured in milliseconds. The task allows practice trials before experimental trials. It has been shown to be a valid tool with an internal consistency of $\alpha = 0.97$ [74].

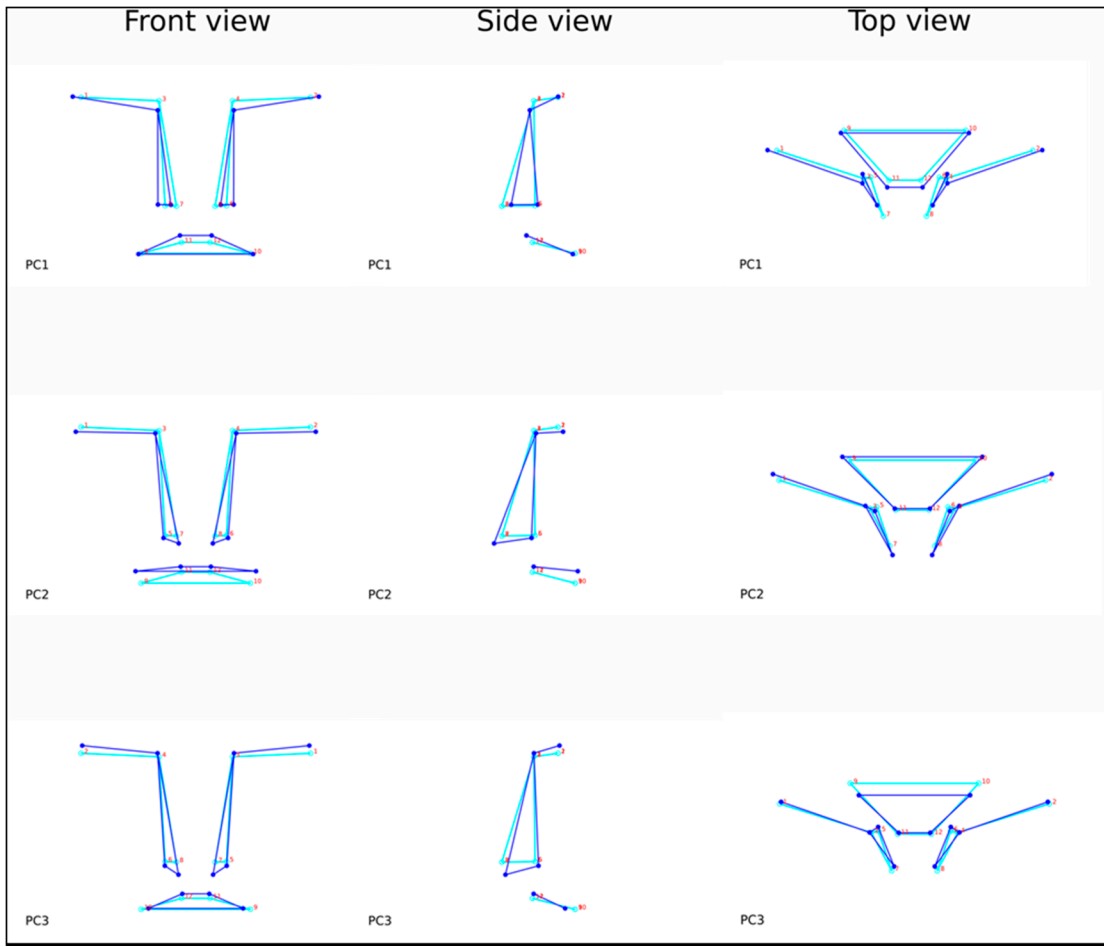

**Figure 2.** First three face shape variation principal components (PCs) observed from the 3D facial scans. Light blue line represents the starting point of the face shape PC (i.e., lower PC score).

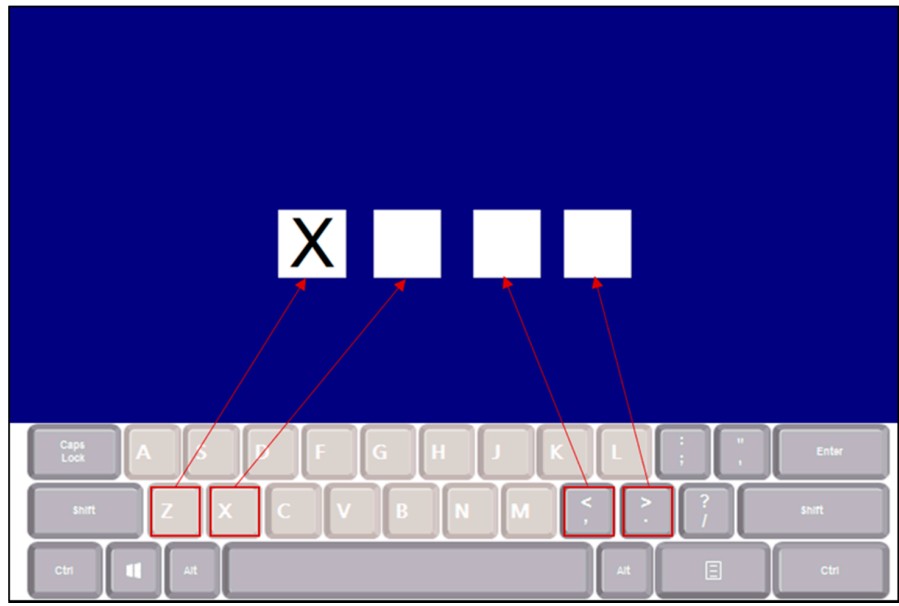

**Figure 3.** Stimulus and response presentation in choice reaction time task. Required responses were 'z', 'x', 'comma', or 'full stop' to a stimulus ('x') emerging in one of the four boxes.

*2.4. Procedure*

Upon collection of demographic information (e.g., age and sex), participants completed the FFQ [61,62] followed by the GPAQ [64] using Qualtrics software with no time restrictions. Participants were instructed to recall their fish product intake and physical activity retrospectively. The CRT test [74] was administered on a standalone PC in the same testing laboratory. Participants were instructed to complete the RT test first—reacting to the same stimulus (an 'x' emerging in a square in the middle of the screen—20 trials in total) by pressing the space bar. Next, participants continued with the CRT task. This time participants were presented with four squares horizontally on the screen. They were asked to react to an 'x' emerging in one of the squares (40 trials in total) by pressing a corresponding key on a keyboard—'z', 'x', 'comma', or 'full stop' (Figure 3). Participants were instructed to place the middle and index fingers of their left hand over the 'z' and 'x' keys and the index and middle fingers of their right hand over the 'comma' and 'full stop' keys. Prior to completing RT and CRT experimental trials, both allowed eight practice trials. The last task was 3D scanning of participants' faces using the Artec MH scanner. Participants wore a swimming cap while instructed to remain motionless with their eyes closed in a seated position.

*2.5. Data Analytic Strategy*

Since age has been shown to have a detrimental impact on CRT [3,12–14] and FA [76,77], we inspected the pattern of zero-order correlations to determine if age was required for covariation purposes in our multiple regression analyses. Three multiple regression models were subsequently developed. The first regression model called "physical system integrity" regressed facial FA on age, self-reported fish product intake, and self-reported physical activity. The second model called "cognitive system integrity", regressed CRT scores on age, self-reported fish product intake, and self-reported physical activity. The third model was a hierarchical multiple regression regressing correct median CRT scores across two steps: (a) Background (sex, age) and Lifestyle Predictors (diet and exercise) were entered on the first step to see if additional variation could be accounted for on the second step called (b) Face Shape Morphological Variation (which included facial asymmetry and data-driven face shape variation principle components PC 1, PC 2 and PC 3. Data are available online in supplementary materials.

## 3. Results

*3.1. Descriptive Statistics*

Table 1 displays the descriptive statistics for the variables involved in the physical and cognitive system integrity models. There were marginal and a statistically significant sex difference in age and median CRT scores respectively. Specifically, men were slightly younger ($t$ (84.71) = 1.96, $p$ = 0.05, $d$ = 0.43), had significantly faster simple ($t$ (86) = 2.16, $p$ = 0.04, $d$ = 0.47) and choice reaction times ($t$ (86) = 2.71, $p$ < 0.01, $d$ = 0.58). Finally, men had higher scores on face shape PC 3 (i.e., thicker lips and downturned noses) compared to women ($t$ (86) = 2.29, $p$ = 0.03, $d$ = 0.49).

The assumption that the data was sampled from a normally distributed population was not met. Specifically, FA and CRT data were positively skewed which is common in the extant research literature. Therefore, the median CRT per subject was used, it is less susceptible to departures from normality compared to the mean [78]. Secondly, FA and CRT data underwent log transformation, after which normality, linearity and homoscedasticity of residuals was met. A univariate and a multivariate outlier were identified. However, their exclusion from the analysis did not change the pattern of findings, thus log transformed data and all analyses included outliers. Likewise, controlling for sex in the system integrity models does not affect the pattern of findings.

**Table 1.** Descriptive Statistics of Study Variables by Sex.

|  | Overall *M* (*SD*) | Male *M* (*SD*) | Female *M* (*SD*) |
|---|---|---|---|
| Age * | 25.74 (9.26) | 23.73 (7.79) | 27.49 (10.13) |
| FFQ | 4.24 (1.26) | 4.43 (1.45) | 4.06 (1.05) |
| GPAQ | 6868.33(7296.45) | 7085.98 (5419.43) | 6678.47 (8664.36) |
| RT * | 288.74(44.99) | 278.24 (29.66) | 297.90(53.66) |
| CRT ** | 436.56 (76.08) | 413.82 (68.80) | 456.40 (77.26) |
| FA (log) | −4.84 (0.26) | −4.80 (0.24) | −4.86 (0.27) |
| DA | 0.0000(0.00003) | 0.0000(0.00003) | 0.0000(.00003) |
| PC 1 | −0.000011(0.0349747) | −0.002929(0.0380788) | 0.002534 (0.0322247) |
| PC 2 | 0.000003(0.0304065) | −0.000041(0.0319512) | 0.000041 (0.0293409) |
| PC 3 * | 0.000026(0.0260132) | 0.006657(0.0286105) | −0.005759(0.0222406) |

Note: * $p = 0.05$; ** $p < 0.01$; *M* = mean; *SD* = standard deviation; GPAQ = Global Physical Activity Questionnaire; FFQ = Food Frequency Questionnaire; RT = Reaction Time; CRT = Choice Reaction Time; FA = Fluctuating Asymmetry Log Transformed; DA = Directional Asymmetry; PC = Principal Component (PC 1 = higher scores indicate wider lips, downturned eyes and flatter nose; PC 2 = higher scores indicate horizontally levelled lips; PC 3 = higher scores indicate downturned nose).

Prior to developing the multiple regression models to test the hypotheses, Pearson's correlations between the predictor and outcome variables were performed to reveal their relationship with the covariate—age. Inspecting the zero-order correlations between age and the study variables (Table 2), it becomes clear that the inclusion of age as a covariate was justified.

**Table 2.** Zero-order Associations among the Study Variables.

|  | FFQ | GPAQ | RT | CRT | FA | DA | PC1 | PC2 | PC3 |
|---|---|---|---|---|---|---|---|---|---|
| **Age** | 0.20 | −0.08 | 0.17 | 0.48 ** | 0.07 | −0.03 | −0.16 | −0.01 | −0.48 ** |
| **FFQ** |  | 0.10 | 0.29 ** | 0.38 ** | 0.05 | 0.01 | 0.05 | 0.007 | 0.07 |
| **GPAQ** |  |  | −0.07 | 0.05 | −0.11 | −0.17 | 0.20 * | 0.03 | 0.07 |
| **RT** |  |  |  | 0.47 ** | 0.16 | 0.14 | 0.01 | −0.08 | −0.20 |
| **CRT** |  |  |  |  | 0.05 | 0.16 | 0.16 | −0.17 | −0.08 |
| **FA** |  |  |  |  |  | 0.07 | −0.06 | 0.23 * | −0.03 |
| **DA** |  |  |  |  |  |  | 0.03 | −0.02 | −0.10 |
| **PC1** |  |  |  |  |  |  |  | 0.00 | −0.00 |
| **PC2** |  |  |  |  |  |  |  |  | 0.00 |

Note: * $p < 0.05$; ** $p < 0.001$; GPAQ = Global Physical Activity Questionnaire; FFQ = Food Frequency Questionnaire; RT = Reaction Time; CRT = Choice Reaction Time; FA = Fluctuating Asymmetry Log Transformed; DA = Directional Asymmetry; PC = Principal Component (PC 1= higher scores indicate wider lips, downturned eyes and flatter nose; PC 2 = higher scores indicate horizontally levelled lips; PC 3= higher scores indicate downturned nose).

Participants who reported on consuming more fish products and those who were older, had worse CRT performance (Figure 4) and lower scores on face shape PC 3 (which is characterized as thicker lips and downturned nose—Figure 2).

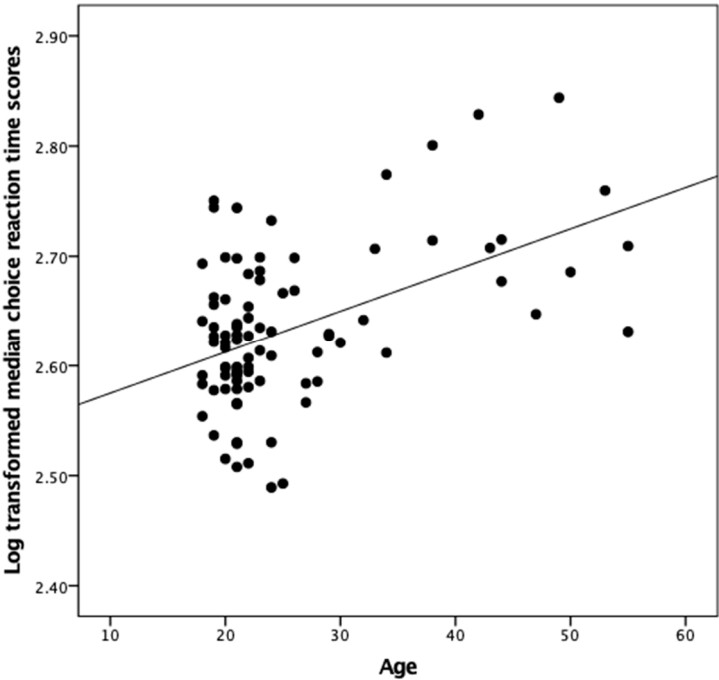

**Figure 4.** Scatterplot represents correlation between age and choice reaction time (CRT).

*3.2. System Integrity Model: What is the Role of Diet and Exercise?*

It was hypothesized that independent of age, more frequent self-reported fish product intake and higher self-reported physical activity would predict higher levels of physical (lower facial FA) and cognitive system integrity (lower median CRT). To test these hypotheses, two multiple regression models were constructed: (a) The physical system integrity model (i.e., outcome variable = log FA) and; (b) cognitive system integrity model (i.e., outcome variable = median CRT). Both models included the following predictors: age, self-reported fish product intake and GPAQ. The third model was a hierarchical multiple regression regressing median CRT scores across two steps in order test if face shape variation including facial fluctuating asymmetry predicts cognitive performance independent of age, diet or exercise. Specifically, the first step included (a) Background (sex, age) and Lifestyle Predictors (diet and exercise) were entered on the first step, then on a second step was included to see if additional variation could be accounted for by (b) Face Shape Morphological Variation (facial asymmetry and data-driven face shape variation principle components PC 1, PC 2 and PC 3.

The physical system integrity multiple regression model was used to fit fish product intake, physical activity and age as predictor variables to uncover their contribution to variability in facial FA. A non-significant overall model emerged ($F$ (3, 84) = 0.49, $p$ = 0.69). Specifically, advancing age, a diet rich in n-3 PUFA, physical activity does not explain statistically significant facial FA variance in this sample.

The cognitive system integrity model was conducted with fish product intake, physical activity and age as predictors and median CRT as the outcome variable. An overall highly significant model was found ($F$ (3, 84) = 13.27, $p$ < 0.001). The model explains 32.1% of the variability in median CRT (Adjusted $R^2$ = 0.30). Table 3 provides with regression coefficients for each of the predictors in the model.

**Table 3.** Cognitive System Integrity Model Regression Coefficients.

|  | *B* | *SE B* | *β* |
|---|---|---|---|
| Constant | 2.47 | 0.03 | |
| Age | 0.003 | 0.001 | 0.43 ** |
| FFQ | 0.02 | 0.005 | 0.29 * |
| GPAQ | 5.68 | 0.00 | 0.06 |

Note: * $p < 0.01$; ** $p < 0.001$; GPAQ = Global Physical Activity Questionnaire; FFQ = Food Frequency Questionnaire.

Table 3 displays that, as expected, age significantly predicted higher scores on CRT test ($β = 0.43$, $t$ (84) = 4.65, $p < 0.001$, partial $r^2 = 0.20$). Older participants performed significantly worse on CRT test than younger participants. Secondly, contrary to the hypothesis, consuming more fish products significantly predicted worsened CRT performance ($β = 0.29$, $t$ (84) = 3.17, $p = 0.002$, partial $r^2 = 0.11$—see Figure 5). Lastly, GPAQ was not a statistically significant predictor of CRT ($β = 0.06$, $t$ (84) = 0.63, $p = 0.63$) in this model.

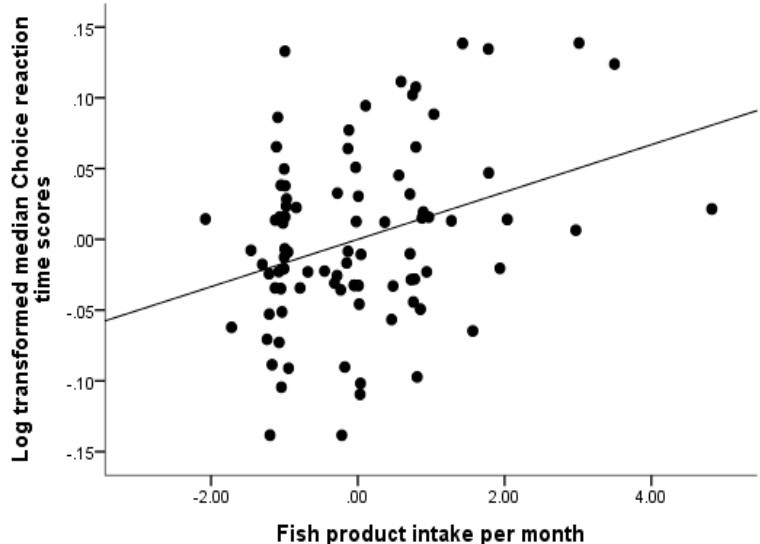

**Figure 5.** Partial scatterplot illustrating best fit line for the significant independent association between higher fish product intake and worse performance on CRT task whilst other predictors are held constant. Both variables are residuals.

*3.3. Does Face Shape and Symmetry Predict Cognitive Performance Above and Beyond Age, Sex, Diet and Exercise?*

A hierarchical multiple regression model regressed median choice reaction time on age, sex, fish product intake (higher scores represent more fish product consumption), physical activity (moderate and vigorous exercise combined where higher values indicate more exercise) on step 1 and then on step 2 facial asymmetry along with the three most important face shape variation components—PC 1, PC 2 and PC 3. Since physical activity was not a statistically significant independent predictor of cognitive performance in the multiple regression ($β = 0.03$, $t$ (79) = 0.34, $p = 0.73$), it was removed from the final model to avoid overfitting [79]. The removal of GPAQ did not change the statistical significance or strength of the remaining significant predictors. The final model was found was highly significant ($F$ (7, 80) = 12.31, $p < 0.001$), it explained 52% of the variability in cognitive performance (Adjusted $R^2 = 0.48$). Regression coefficients for each of the predictors in the two-step model are presented in Table 4. Furthermore, the addition of the second step of facial morphology factors resulted in statistically significant change in variation ($R^2$ change = 0.12, $p = 0.002$) above and beyond lifestyle (i.e., diet) and background factors (i.e., age, sex).

**Table 4.** Hierarchical Multiple Regression Coefficients for the Predictors of Median Choice Reaction Time (i.e., higher scores indicate poorer performance). First Step is Testing for Lifestyle and Background Effects and the Second Step is Testing if Facial Morphology and Developmental Stability Independently Predicts and Accounts for a Significant Change in the Variance in Cognitive Performance.

| | *B* | *SE B* | *β* | *Partial r²* |
|---|---|---|---|---|
| Lifestyle and background factors ($F(3,84)=18.93$, $p < 0.001$, Adjusted $R^2 = 0.38$) | | | | |
| Constant | 204.61 | 33.35 | | |
| Age | 3.08 | 0.73 | 0.38 ** | 0.24 |
| Sex | 39.19 | 13.32 | 0.26 ** | 0.12 |
| FFQ | 21.82 | 5.30 | 0.36 ** | 0.15 |
| Facial morphology factors ($F(7,80)=12.31$, $p < 0.001$, Adjusted $R^2 = 0.48$) | | | | |
| Constant | 169.12 | 33.13 | | |
| Facial asymmetry | 57,5746.81 | 267,024.74 | 0.17 * | 0.06 |
| PC 1 | 456.66 | 174.51 | 0.21 * | 0.08 |
| PC 2 | −399.94 | 194.14 | −0.16 * | 0.05 |
| PC 3 | 588.60 | 266.44 | 0.20 * | 0.06 |

Notes: * $p < 0.05$; ** $p < 0.01$; FFQ = Food Frequency Questionnaire; PC = Principal Component (PC 1 = higher scores indicate wider lips, downturned eyes and flatter nose; PC 2 = higher scores indicate horizontally levelled lips; PC 3 = higher scores indicate downturned nose).

In the final step, older participants ($β = 0.38$, $t(80) = 4.24$, $p < 0.001$) and women ($β = 0.26$, $t(80) = 2.94$, $p = 0.004$) had worse cognitive performance (i.e., slower CRT = higher scores). Contrary to our expectations, fish product intake was a significant predictor of worsened cognitive performance ($β = 0.36$, $t(80) = 4.12$, $p < 0.001$). Facial asymmetry ($β = 0.17$, $t(80) = 2.16$, $p = 0.034$), face shape variation PC 1 ($β = 0.21$, $t(80) = 2.61$, $p = 0.011$), PC 2 ($β = −0.16$, $t(80) = −2.06$, $p = 0.043$), and PC 3 ($β = 0.20$, $t(80) = 2.21$, $p = 0.030$) predicted cognitive performance. Specifically, increased facial fluctuating asymmetry and higher scores on face shape components PC 1 and PC 3 (lower scores on PC 2) predicted worsened or slower CRT scores. Specifically, people with symmetrical faces, more horizontally levelled orbital apertures, horizontally levelled lips, less downturned noses and thinner, broader lips (Figure 2) had better cognitive performance.

## 4. Discussion

Facial symmetry, independent of sex, age, diet and exercise, was a positive predictor of cognitive performance (i.e., lower CRT scores). Controlling for age was particularly important given ours and previous research [3,13,14], showing that advancing age predicted worsened CRT scores. Surprisingly, there was an unexpected statistically significant positive association between having a diet rich in n-3 PUFA (acquired from the consumption of fish products including fish oils) and worsened CRT performance (i.e., higher CRT scores). Contrary to expectations, there was no statistically significant association between self-reported physical activity and cognitive performance or FA in this sample. Importantly, independent of diet (or exercise), facial symmetry and multiple face shape components predicted better cognitive performance and could ameliorate age-related or early cognitive decline. The latter conjecture is speculative given our cross-sectional findings as without longitudinal data we cannot be certain a participant's baseline or cognitive performance when they were younger. Regardless, it is important to note that individuals with horizontally levelled orbital apertures, less downturned noses and thinner, broader lips had better choice reaction time performance (adjusted for chronological age). The underlying causes of the associations between facial morphology and cognitive performance are not known. However, previous work on developmental integrity and cognitive performance and age-related declines in system integrity are worth exploring in greater detail [3,11].

Deary [4] proposed that the latent variable "developmental integrity" explains the commonly reported associations between cognitive and physical performance. If correct, one would expect that system integrity declines with advancing age due to an accumulation of genetic and environmental

perturbations throughout one's lifespan [5,14]. Neurobiologically, information-processing slows with age due to axonal demyelination and loss of white matter that occurs as individuals become older [80]. Reduction of white matter is one of the characteristics of an ageing brain [81] and has been directly linked to decreased processing speed [82]. In addition, the quality of information-processing reduces with age [3,83]. Response production [13] is likely affected by ageing via decline in motor control including difficulties coordinating movements and managing multiple cues. These changes have similarly been associated with white matter changes in the corpus collosum, as it contains connections between the hemispheres and directs bimanual coordination likely to be important in CRT performance [84].

It is widely known that cognitive systems decline in middle age [85], however, there is evidence suggesting that decline begins in healthy, educated adults in their 20s and 30s shortly after maturation [86]. Thompson, Blair and Henrey [1] showed that decline in performance can begin as early as when an individual is 24 years old. These findings are consistent with the steady cross-sectional decline in information processing speed observed in the sample here. It was also observed that there was little increase in CRT among individuals aged 18–30 but a substantial increase among those above 30 years of age, suggesting that cognitive decline might not be a linear process and could be accelerating as a person gets older. Surprisingly, our findings indicate that cognitive performance and possibly early cognitive decline could be explained, in part, by individual differences in underlying developmental stability independent of diet or exercise. Despite n-3 PUFA and aerobic exercise interventions studies reporting beneficial outcomes [27,35,38,40], this benefit was not reflected in the current findings. This is surprising given the fact that physical activity induces neurocognitive changes most commonly in frontal and hippocampal regions [87–89]. Below we explore why our measures of diet and exercise did not have the expected associations on cognitive performance.

Rosano et al. [90] found greater activation in frontal and temporal regions during an attention task among physically active older adults. However, the cognitive task used by Rosano et al. [90] involved perceptual organization and selective attention which differs from the CRT task [74] we used here. This methodological difference could explain the lack of relation between exercise exposure and CRT. Another reason that physical activity did not have an association with any of the study variables here could be measurement error in GPAQ [64]. The questionnaire does not have high construct validity or test-retest reliability [65]. Participants were required to indicate the number of hours, minutes and days being physically active and were unrestricted when specifying. In some cases, responses exceeded credible time that one could be realistically physically active. This suggests that some people may overestimate their physical activity. Previous work indicates that self-report physical activity questionnaires can include faults in memory recollection [91], inconsistency of physical activity over time [92] and social desirability leading participants to over-report their activity [93]. Moreover, a systematic review of validity and reliability studies on self-reported physical activity identified an overall low methodological quality and absence of an ideal self-report physical activity measure [94].

In contrast, responses on the FFQ measuring seafood intake [61,62] might be a more accurate measure of actual behavior as people do not hesitate reporting about fish product aversions [95]. The association between increased fish product consumption (including fish oils) and worsened choice reaction time is intriguing given that n-3 PUFA was likely important for healthy neural development [32,96–98]. For example, n-3 PUFA deficiency in the brain has been linked to several human mental and neurological disorders [99]. Subsequently, it is not a surprise that n-3 PUFA supplement intervention studies have repeatedly shown to improve cognitive functioning and structural brain health [25,27–29] to reduce cognitive decline in healthy individuals [30,31] and have some effect in counteracting AD symptoms in its onset stage [26]. One reason that our findings may not be consistent with the extant literature could be the existence of an underlying problem when self-reporting a single dietary component in isolation. Consuming fish products does not necessarily lead to consuming an overall healthy diet. A more detailed FFQ would enable estimating if n-3 PUFA impact is not being neutralized with other foods consumed. For example, intake of a diet rich in trans

and saturated fat has been associated with cognitive deficits [100], general caloric overconsumption has shown to reduce synaptic plasticity [101] and to exacerbate cognitive decline [102]. Namely, an overall diet of an individual would need to be considered to evaluate if amount of n-3 PUFA could be beneficial given other foods consumed.

Importantly, however, there are contradictory findings regarding the beneficial impacts of n-3 PUFA on neurocognitive function. For example, Köbe et al. [40] found increase in grey matter in AD associated frontal region only when n-3 PUFA supplement was combined with physical activity. The supplement group did not show any increase in brain volume. Howe, Evans, Kuszewski and Wong [103] administered fish oil supplement in a randomized control trial for 20 weeks in a group of older adults. There was some structural change observed—cerebrovascular responsiveness to hypercapnia, but only in women. No associated changes in cognitive performance were found. Andrieu et al. [104] compared n-3 PUFA supplement to a placebo group to evaluate for differences in cognitive decline over three years, but no significant differences were found. Inconsistent findings in n-3 PUFA intervention studies suggest that there might be individual differences in susceptibility to related cognitive and health improvements. Therefore, to detect if either n-3 PUFA or physical activity have beneficial effects on age-related cognitive decline, it would best to know the initial state of developmental integrity for each subject especially if such individual differences predict responsiveness to lifestyle interventions.

It is worth considering that the beneficial effects of diet or exercise may exhibit a dose-response relation. Specifically, the in-utero effects of maternal diet and exercise likely have an inverted U dose-response relation on offspring epigenetics [105,106]. Maternal exercise is beneficial on fetal development at low to moderate levels, but not at high levels. It remains to be seen if in utero exposure to beneficial diets or exercise shape subsequent adult responses. Beyond inverted U exercise epigenetic responses proposed by Chalk and Brown [106], de Groot, Ouwehand and Jolles [107] and Chang, Labban, Gapin, and Etnier [108] have reported inverted U-shaped associations with regards to cognitive performance for fish consumption and exercise respectively. Therefore, a reconceptualization may be needed regarding the non-linear hormesis dose-response relation [106] between diet, exercise, and cognitive performance. Generally, the idea is that moderate amounts of exercise or dietary stress is beneficial, but at high levels both are detrimental to development.

In terms of intervention responses, individuals may not equally benefit from diet and exercise exposure. Indeed, developmentally stable individuals may need lower doses to achieve high benefits much like receptor sensitivity in neural systems. Likewise, individuals of lower developmental integrity may require higher dietary doses to receive the equivalent benefits. Thus, some individuals might be more responsive to lifestyle changes than others. Our findings show that cognitive performance was explained by various theoretically driven (i.e., FA) and data-driven measures of face shape variation. Thus, human faces, besides serving as a tool in social interactions, may also project the loss of phenotypic quality or system integrity of the organism. It should not be surprising that face shape variation appeared to be better predictor of cognitive performance than facial asymmetry. The latter is prone to measurement error and biased by directional asymmetries in facial muscle and expression. Likewise, Scheib, Gangestad and Thornhill [109] showed that humans are capable of detecting 'good genes' even when cues to facial asymmetry are unavailable. Therefore, in addition to providing a sign for phenotypic quality, human faces could indicate if an individual is genetically predisposed to healthy ageing.

There are instantly visible age-related changes to skin texture of the face caused by senescence, such as increased FA [110], loss of facial fat/muscle [53] and changes to bone density [54]. It seems that midface and orbital areas are inclined to substantial transformation with age—orbital apertures re-shape and pyriform aperture widens [54]. Furthermore, Gunn et al. [111] showed how individuals of the same chronological age can differ in their perceived age, whilst Okada et al. [50] demonstrated how smoking twins can appear older than their non-smoking counterparts, suggesting that individuals vary in the way they physically age. In our study, our data-driven face shape variation components were

associated with cognitive performance when age was controlled. That is, individuals who had more horizontally levelled orbital apertures had better cognitive performance. There was also variability in the midface observed - individuals with more downturned noses had worse cognitive performance. However, it was individuals with thinner, wider lips who had increased cognitive performance when age is controlled. Some of our face shape components (e.g., PC3 correlates with age in our sample) or are similar to previous facial morphological indicators of older age such as thinner lips [53]. Thus, it is important that our face shape components are independently associated with cognitive performance when age is controlled. Nonetheless, a more complete investigation of facial components longitudinally could provide a better insight of which cues could be signifying healthier cognitive ageing. In addition, our morphological measurement was limited to 12 manually placed facials points. A more spatially dense analysis of the 3D facial scans could provide richer data to see their associations with general decline seen in human faces as found in previous studies [53,54]. As for example, computer algorithm developed by Ekrami et al. [112] showed an accurate estimation of FA across the entire face and it could be implemented in future studies.

Individual differences in facial ageing are unexplained [50,111] as are those found in cognitive ageing [113]. Some individuals seem to have younger cognitive and perceived age despite their chronological age and an underlying trait of system integrity postulated by Deary [4] could hold an answer to this 'hidden' covariation. Identifying factors that account for individual variability in cognitive and physical ageing could potentially introduce ways to counteract ageing effects (e.g., improving the brain's overall processing efficiency) and to increase a healthy lifespan. Those who exhibit more signs of early ageing could benefit from interventions that target decline before its onset. The spread of age in our sample was good (i.e., 18 to 55 years of age), but with a tendency for younger mode typical of university students. Future research is needed to extend our findings to a wider population. This is critical considering that the possibility that mature university students represent a unique population whose cognitive skills are more consistent with a younger age group because of their ongoing learning.

## 5. Conclusions

To conclude, the current findings provide evidence that enhanced cognitive performance is associated with developmental stability independent of age and lifestyle factors (e.g., diet and exercise). Specifically, independent of sex, age, diet or exercise, face shape and symmetry—commonly used measures of developmental quality—predicted cognitive performance. Surprisingly, n-3 PUFA intake was associated with worsened cognitive performance, whilst physical activity was unrelated to cognitive performance or symmetry in this sample. Such findings suggest that some individuals' neurocognitive systems age better than others despite life style factors. Longitudinal work is needed on the environmental and developmental predictors of cognitive performance, so we may be able to ameliorate the negative effects of adverse lifestyles or those at increased risk of early cognitive decline.

**Supplementary Materials:** Data are available online at: https://osf.io/m7n36/?view_only=b7e51060714543bea72f313eca7c4a61.

**Author Contributions:** Conceptualization, W.M.B. and A.U.; methodology, W.M.B. and A.U.; software, W.M.B.; validation, A.U.; formal analysis, W.M.B.; investigation, A.U.; resources, W.M.B.; data curation, W.M.B.; writing—original draft preparation, A.U. and W.M.B.; writing—review and editing, A.U. and W.M.B.; visualization, A.U. and W.M.B. supervision, W.M.B.; project administration, W.M.B.; funding acquisition, W.M.B.

**Funding:** This research received no external funding.

**Conflicts of Interest:** The authors declare no conflict of interest.

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
