# Peer review of "The Face of Early Cognitive Decline? Shape and Asymmetry Predict Choice Reaction Time Independent of Age, Diet or Exercise"

_symmetry, doi:10.3390/sym11111364_

Round 1

Reviewer 1 Report

The authors present results from a study relating reaction time to facial shape and asymmetry (FA), as well as diet and physical activity. Since age is an important factor affecting reaction time, an attempt is made to correct for age effects. Overall the paper is very well written and makes reference to a very broad literature. There are some aspects which I think can be improved:

1) The authors often refer to CRT decline, while they actually measure CRT at a specific age. From the scatter plot of CRT and age it can be clearly seen that indeed none of the older (40+) participants have a CRT below 400, but still one has no idea from which level they came when they were 20. I agree this is just a detail, but it is important to note that the authors do not measure decline, but rather CRT.

2) I am not a big fan of residual analyses. Correcting CRT for age with a linear regression cleary assumes linearity. I am absolutely convinces that the decline in CRT with age accelerates as one gets older. As can be seen from figure 4, there is hardly any change in CRT with age in the 18-30 years group of participants, but then after the age of 30 the increase in CRT emerges. I think the analysis as such can be kept in the manuscript, but some more caution may be added.

3) related to my previous suggestion, I think it would be interesting to test for an association between CRT and FA for the 18-30 years group. The effect of age in that group can likely be neglected. So if some type of stress would both affect CRT and facial FA, it should emerge in this subgroup.

4) why is median CRT not log-transformed in figure 4 while it is in figure 5?

5) In table 2, correlations are presented for DA (directional asymmetry), while in the analyses presented here, DA is considered as a population average (as it is done in most papers). Please clarify what DA means in this table.

6) Although there is nothing wrong with using a limited number of landmarks, I find it a pity that from a facial 3D scan only very limited information is captured. There is a growing literature on how to take full advantage of the full 3D scan instead of putting manual landmarks for a limited number of facials points. I suggest the authors to read a recent paper by Ekrami et al. (2018) in PlosOne (https://journals.plos.org/plosone/article?id=10.1371/journal.pone.0207895) to get more inspiration and get much more out of their data.

Author Response

Please see attached our response to Reviewer 1.

Reviewer 2 Report

This manuscript detailed the associations between cognitive decline (as measured by choice reaction time) and facial symmetry measures, exercise, and dietary intake of fish (due to the importance of a number of dietary compounds in fish that have been shown to positively influence brain development and protect against cognitive decline). The authors used a range of robust measures of facial symmetry, exercise, and dietary intake, and proposed several sensible theoretical models that tested these associations. They find that age, exercise, and fish intake do not predict facial symmetry (the physical system model) but that age, exercise, and fish intake do predict measures of choice reaction time (the cognitive system integrity model), or rather the other measures do while exercise is uninformative. An exploratory model found that age, sex, diet, and measures of facial symmetry are able to predict age related cognitive decline; a measure derived from the residuals of the association between age and choice reaction time. Unusually, the authors find that while facial symmetry does predict this outcome, more fish consumption is associated with worse performance.

The manuscript was exceptionally well written, researched, and carried out - I found the background and motivation to be well justified and the model specifications mostly clear. I appreciated the authors careful reporting of statistics, design decisions, and making the data open. I don’t have many comments about the manuscript in general, and thanks to the data being made available, was able to recreate the analyses reported in the paper successfully. This also allowed me to check what appeared to be an anomaly in Table 4, which showed an misspecification in the model that means the conclusions of the paper should change somewhat.

Table 4 reports the results of the exploratory model between the residual measure and the age, sex, diet, and symmetry measures, and has an excellent model fit. This seemed very high to me, and noted that the standardised coefficient of Age is .88, by far and away the most influential of all predictors in this model. This caused concern for one reason - the residuals the authors specify as the outcome variable state that ‘Higher observed scores on the residuals indicated a better CRT performance than expected given a person’s age’ implies that the Age predictor must be uncorrelated with the residuals. Because the residuals represent the error of the predictions by using Age to predict Correct Median CRT, they are orthogonal to Age (from the geometric interpretation of linear regression).

By inspecting the data I found that the residuals were back-to-front - they are the errors that stem from predicting Age from Correct Median CRT, rather than the other way around. That is, higher residual scores represent a person being older than you would guess by knowing their CRT score and vice versa, which is clearly different from what was intended. Though the correlation between Age and CRT is unchanged whichever way you specify them, treating one as a outcome and one as a predictor results in a different least squares estimation and thus changes the residuals (this is because least squares is a form of vector projection and the vectors are of different lengths; thus projecting A onto B is not the same as B onto A, for example). This is why Age is such a strong predictor in the original model - it is of course able to predict a slightly modified version of itself. 

By specifying the residuals in the intended manner, the new model shows a much reduced, yet still significant, model fit: Adj R2 = 0.30, F(7, 80) = 6.39, p < .001. Here, Age is null predictor as would be expected (it is orthogonal), females have significantly higher scores than males (females being the reference category) greater absolute DA plus FA is associated with higher residual scores, and PC’s 1, 2, and 3 are significantly related also in varying directions. Interestingly, the ‘fish intake sum lower better’ variable (I am assuming this is the correct variable as I was able to recreate all other analyses with it) showed a negative relationship with the residuals. This is a brain-twister, but I think this means a lower (better) fish intake score means a higher residual score; i.e., someone doing better for their age than expected on the CRT, which is more in-line with the literature in the introduction.

The above re-analysis changes the interpretation of the data, so the discussion will need to be reworked to reflect the fact that greater asymmetry seems to predict better performance as does greater intake of fish.

Finally, there was an additional point I wanted to add around the model specifications used - in the first two models, the log of the dependent variables was used, whereas to compute the residuals, the median CRT was used. One of the assumptions of linear regression is not that the variables themselves should be normally distributed, but rather that the residuals should be normally distributed around zero. For the cognitive system integrity model, this assumption was met using Median CRT, so it is more expressive and interpretable to use that variable. For the physical system model the authors are right to use the log transform as the non-logged FA measure does not have normally distributed errors. The final point, then, is to make the final model simpler in its interpretation - rather than computing residuals, why not take a straightforward model comparison approach? If the authors are interested in how facial symmetry measures are associated with CRT independent of diet or exercise (as stated in the abstract) then the authors can test that directly. For example, if we create a base model thus:

Base = Median CRT ~ Age + Sex + Fish Intake + Vigorous exercise

Then facial symmetry measures can be added, and the change in variance explained tested for significance:

Symmetry model = Base + Absolute DA + FA + PC1 + PC2 + PC3

Using this approach the authors should find symmetry measures explain a significant further 11% of the variance in median CRT, independent of age, sex, diet, and exercise.

Author Response

Please see attached our response to Reviewer 2.

Reviewer 3 Report

I have appended my review to the Word document attached. On the front end, I can say that this is certainly an interesting topic but I am concerned about the low statistical power to draw these inferences. The authors need to take great care to justify the veracity of their findings.

Author Response

Please see attached our response to Reviewer 3.

Round 2

Reviewer 2 Report

I would like to thank the authors for their consideration of my comments, and adoption of the suggestions I made for the modelling strategy. Reading over the manuscript, I find it much improved, but still have some serious reservations about the statistical analysis, which come in two points below.

The authors have retained their analysis of the residuals at the request of another reviewer. Of course, a response to reviews is a balancing act between the different suggestions reviewers make. However, in this case, the authors have ignored the point I made about the residual analysis - that it is mathematically wrong. The manuscript still reports the model fit as being exceptionally high, with age as a significant predictor. This is incorrect, as the residuals, the outcome measure, have been miscalculated. As I said before, the way to do this in the way the authors have suggested is to predict correct median CRT from age, collect the residuals, and then use those as the dependent variable in the regression as stated - when this happens, the correct model outcome if F(7, 80) = 6.39, p < .001, adjR2 = .302, and the predictors all have different values that are not what appear in the table. At this point, I would request the authors remove this analysis - it is wrong and is subsumed by the current strategy; and so does not pose any use to future researchers conducting longitudinal studies on symmetry. The second reservation I have is that while the authors have adopted the modelling strategy suggested, the statistics are heavily misreported. The statistics reported in section 3.4 and Table 4 are almost all incorrect. For example; the final model, of the form: Correct Median CRT ~ Age + Sex + FishIntake + Exercise + Asymmetry + PC1 + PC2 + PC3 has the following fit, F(8, 79) = 10.7, p < .001, adjR= .47. The variance explained statistics are right, but the F is not, in the paragraph of 3.4 In the table, the model fits of both the first step (CRT ~ Age + Sex + Fish + Exercise) is off by a few points (F(4, 83) = 14.2, p < .001, adjR2 = .38, and the coefficients do not match at all. The same is true for the results of when the facial morphology scores are included for both the model fit as well as the coefficients. In the discussion of the final model following table 4, the coefficients and t and p values are all also different. These need to be corrected - I am not sure what is causing the difference here, as I am working from the data source provided in the initial round of reviews.

Author Response

We would like to give our sincere thanks for the great assistance and comments regarding our manuscript. Please see attached our responses.

Sincerely,

Dr William M Brown and Agnese Usacka

Reviewer 3 Report

The authors addressed my concerns adequately.

Author Response

Dear Reviewer 3,

Thank you very much for your assistance and guidance during this process.

Sincerely,

Dr William M Brown and Agnese Usacka